# Alleviation of Ulcerative Colitis Potentially through th1/th2 Cytokine Balance by a Mixture of Rg3-enriched Korean Red Ginseng Extract and *Persicaria tinctoria*

**DOI:** 10.3390/molecules25225230

**Published:** 2020-11-10

**Authors:** Evelyn Saba, Yuan Yee Lee, Man Hee Rhee, Sung-Dae Kim

**Affiliations:** 1Department of Veterinary Biomedical Sciences, Faculty of Veterinary and Animal Sciences, Pir Mehr Ali Shah Arid Agriculture University, Rawalpindi 46000, Pakistan; evelyn.saba@uaar.edu.pk; 2Department of Veterinary Medicine, College of Veterinary Medicine, Kyungpook National University, Daegu 41566, Korea; yuanyeelee@knu.ac.kr; 3Research Center, Dongnam Institute of Radiological and Medical Sciences, Busan 46033, Korea

**Keywords:** Rg3-RGE, *Persicaria tinctoria*, ulcerative colitis, *NF-κB*, *NLRP3*

## Abstract

Ginseng is a vastly used herbal supplement in Southeast Asian countries. Red ginseng extract enriched with Rg3 (Rg3-RGE) is a formula that has been extensively studied owing to its various biological properties. *Persicaria tinctoria* (PT), belonging to the *Polygonaceae* family, has also been reported for its anti-inflammatory properties. Ulcerative colitis (UC) is inflammation of the large intestine, particularly in the colon. This disease is increasingly common and has high probability of relapse. We investigated, separately and in combination, the effects of Rg3-RGE and PT using murine exemplary of UC induced by DSS (Dextran Sulfate Sodium). For in vitro and in vivo experiments, nitric oxide assay, qRT-Polymerase Chain Reaction (PCR), Western blot, ulcerative colitis introduced by DSS, Enzyme Linked Immunosorbent Assay (ELISA), and flow cytometry analysis were performed. The results obtained demonstrate that treatment with Rg3-RGE + PT showed synergism to suppress inflammation (in vitro) in RAW 264.7 cells via mitogen-activated protein kinase and nuclear factor κB pathways. Moreover, in C57BL/6 mice, this mixture exhibits strong anti-inflammatory effects in restoring colon length, histopathological damage, pro-inflammatory mediators, and cytokines amount, and decreasing levels of *NLRP3* inflammasome (in vivo). Our results recommend that this mixture can be used for the prevention of UC as a prophylactic/therapeutic supplement.

## 1. Introduction

Inflammatory bowel disease (IBD) is a term that describes two intestinal abnormalities, ulcerative colitis (UC) and Crohn’s disease (CD). Severe abdominal discomfort including hematochezia, diarrhea, and pain are characteristic manifestations of IBD. In the past few years, this disease has attracted great attention especially in developed countries, with reports indicating that greater than 1 million humans suffered from IBD in the US only [1,2]. An inefficient intestinal epithelial barrier and imbalance in the commensal microflora are responsible for IBD disease symptoms. Moreover, microscopically, IBD is demonstrated by extensive movement of neutrophils, monocytes and many types of lymphocytes in the intestinal epithelium which in turn release pro-inflammatory mediators and cytokines that aggravate inflammation. If left unresolved, chronic epithelial damage and even colorectal cancer (CRC) may occur [3].

UC, a form of IBD, occurs as a consequence of imbalance of T-helper (Th) cell types 1 and 2. Previous data have shown that a shift of balance toward Th2 cell cytokine production occurs during UC, whereas in CD, there is a bias toward Th1 cell cytokine production [4]. To date, the exact mechanism for the development of UC and CD is yet to be elucidated since the etiological factors responsible for both IBD forms are numerous; they range from the intestinal microenvironment to the general outer macro-environment and genetic susceptibility [5]. Inflammasomes are multiprotein oligomers that are responsible for producing pro-inflammatory cytokines. Nod-like receptor family pyrin domain-1 containing 3 (*NLRP3*) are the pattern-recognition receptors (PRR) that constitute inflammasomes, with a major part in the development of colitis via secreting pro-inflammatory cytokines [6,7].

In the Korean peninsula, *Panax ginseng* is used extensively as an herbal supplement for its therapeutic and preventive effects. Its name *Panax* has been given to ginseng because it is derived from “panacea” which means cure for all; this also includes the beneficial effects of ginseng in almost all disease conditions [8,9]. The active components called ginsenosides have biological activities of ginseng. To date, numerous ginsenosides have been discovered and used in experiments due to their specific biological activities [10]. Among them, Rg3 is a top-priority ginsenoside as it elicits strong biological properties in many ailments including hypertension, inflammation, and cancers [11,12]. Another beneficial herb, *Persicaria tinctoria* (PT), is known for its anti-inflammatory and antioxidant characteristics [13].

To date, however, a study examining the effects of Rg3-RGE and PT on intestinal disease such as colitis has not been performed. Therefore, we researched to find the combined effects of Rg3-RGE and PT on the colitis induced by DSS in a murine model.

## 2. Results

### 2.1. Bioactive Compounds in Rg3-RGE and PT

We investigated the bioactive compounds of Rg3-RGE via HPLC analysis and PT via UPLC–QTofMS analysis. The chromatogram for Rg3-RGE is shown in Figure 1A, with peaks labeled with the corresponding ginsenosides. The chromatogram for PT is shown in Figure 1B and the information for each peak is summarized in Table 1.

### 2.2. Rg3-RGE + PT Synergistically Attenuated LPS-Induced Inflammation

The basic endotoxin from the outer layer of Gram-negative bacteria, lipopolysaccharide (LPS), evokes a variety of inflammatory responses such as nitric oxide (NO) production, increased levels of arachidonic acid, and the stimulation and release of cytokines and chemokines. NO production is regarded as the defensive action of cells toward endotoxic shock [14]; however, its excessive stimulation can lead to cellular death, implying that NO levels should be controlled to prevent cytotoxicity. In this experiment, it is demonstrated that Rg3-RGE + PT showed synergism and faded the levels of NO in RAW 264.7 cells with no cytotoxicity (Figure 2A,B).

### 2.3. Effects of Rg3-RGE + PT on the Expression of Pro-Inflammatory Mediators and Cytokines at the Transcriptional and Translational Levels

The expression of pro-inflammatory mediators and cytokines are found to serve as a major response to the biological system to inflammation [15]. That is why we determined whether Rg3-RGE + PT in RAW 264.7 cells could affect the pro-inflammatory cytokine expression. Synergism with Rg3-RGE + PT significantly abolished the expression of *iNOS*, *COX-2*, *IL-1β*, *IL-6* and *TNF-α* mRNA as shown in Figure 3A,B. In addition, *iNOS* and *COX-2* translational levels were inhibited (as shown in Figure 3C,D).

### 2.4. Signal Transduction of Rg3-RGE + PT via NF-κB and Mitogen-Activated Protein Kinase (MAPK) Pathways

In inflammatory signaling, *NF-κB* serves as a top-priority pathway which is responsible for a vast array of transcription factors and genes associated with immunity, stress, and inflammation. When the ligand–receptor interaction occurs, transforming growth factor beta activated kinase 1 (*TAK1*) phosphorylation results in downstream activation of the *TAK1* factor. The phosphorylated *TAK1* then further travels downstream and activates *IKKα/β* resulting in its phosphorylation and *IκBα* activation. Phosphorylated *IκBα* then frees *NF-κB* which then translocates to the nucleus, initiating the inflammatory pathway [16]. Evidently from Figure 4A,B, Rg3-RGE + PT in macrophages inhibited phosphorylation of *NF-κB* along all its upstream factors in vitro.

The pathway of mitogen-activated protein kinase (*MAPK*) is another crucial pathway for cell response to stress and is co-stimulated to evoke an inflammatory response with *NF-κB* pathway [17]. As with the *NF-κB* pathway, it has several factors whose subsequent inflammation is mediated by phosphorylation. Therefore, we evaluated the effects of Rg3-RGE + PT on the level of expressions of the factors involved in the *MAPK* pathway. As shown in Figure 4C,D, Rg3-RGE + PT inhibited the phosphorylation of both upstream and downstream factors, meaning we had produced a potent anti-inflammatory mixture.

### 2.5. Preventive Role of Rg3-RGE + PT in DSS-Induced Colitis

Investigating the preventive influence of Rg3-RGE + PT on colitic mice caused by DSS, we treated mice with the two extracts alone and combined, and with a positive control (sulfasalazine) for 7 days. After 7 days, colon was extracted, and its length measured as shown in Figure 5A,B. Normal length was retained in the mixture-treated group; however, Rg3-RGE, PT, and the mixture recovered the weight of mice, which severely decreased due to DSS treatment as shown in Figure 5C. Disease activity index (DAI) was also found to be efficiently recovered in the mixture group as shown in Figure 5D. In addition, the histological damage as loss of epithelial cell integrity, thickening of the mucosal wall and infiltration of inflammatory cells in the colon tissue, as clearly observed in the DSS community, were reversed positively by treatment with the mixture (Figure 5E). Histopathological lesions in the colon tissue were classified according to the scoring method given by Geoboes et al. [18] as shown in Figure 5F.

### 2.6. Effects on Cytokine Production in DSS-Induced Colitis Mice

To investigate the expression of cytokines in mice model, plasma was extracted from DSS-induced colitis mice and level of expression of pro-inflammatory mediators and cytokines (*NO*, *IL-1β*, *IL-5*, *IL-13*, and *TNF-α*) were verified. The positive control and mixture group significantly inhibited these levels when compared to the DSS group as shown in Figure 6A–E. A radar chart was plotted and it was shown that the combination of Rg3-RGE and PT is more potent in suppressing NO, *TNF-α* and *IL-1β* (Figure 6F).

### 2.7. Transcriptional Suppression of Pro-Inflammatory Mediators and Cytokines and Signal Transduction of Rg3-RGE + PT via NF-κB in Colon Tissue

To verify that RNA obtained from DSS-induced mice colon exhibited the same expression as macrophage cells, mRNA, and protein expression were demonstrated for *iNOS*, *COX-2*, *IL-1β*, *IL-6* and *TNF-α* and *NLRP3* (Figure 7A,B), the positive control and mixture-treated group had significant inhibition in pro-inflammatory mediators and cytokine expression at the transcriptional level. At the translational level (Figure 7C), *NLRP3* and *NF-κB* inhibition strengthen our hypothesis that the mixture has targeted the *NF-κB* pathway for alleviation of inflammation.

### 2.8. Effects of Rg3-RGE + PT on Immune Cell Subtypes in Spleen

DSS decreased the number of CD4^+^ T-cells and regulatory T-cells (Tregs) as shown in Figure 8A. However, treatment with sulfasalazine (positive control), Rg3-RGE, and PT recovered the number of CD4^+^ T-cells and Treg cells; the number of B-cell subtypes remained the same as shown in Figure 8B. Samples treated with the mixture had a significant rise in the number of Treg cells (Figure 8A) and CD4^+^ T-cells (Figure 8C) which confirms its synergistic effect compared to individual treatment of Rg3-RGE and PT (Figure 8). Since CD4^+^ T-cells were highly elevated, we sought to investigate the differentiation of T-helper cells by examining secretion of *IFN-γ* related to Th1, and *IL-4* related to Th2 in anti-CD3 stimulated splenocytes. As shown in Figure 9A, *IFN-γ* was elevated by treatment with DSS, and reduced by treatment with sulfasalazine, Rg3-RGE, and PT. As expected, the mixture of Rg3-RGE and PT strongly reversed *IFN-γ* secretion when compared with the control group, which indicates that differentiation of naïve CD4^+^ T-cells to Th1 cells had decreased; however, differentiation to Th2 cells increased (owing to increased secretion of *IL-4*; (Figure 9B)). However, we should perform T-cell negative purification to remove all other T-cells such as the Natural Killer (NK) cells which may also secrete *IFN-γ* for a more accurate detection of Th1 related *IFN-γ* detection. From our findings, there may be a possibility that the mixture enhanced the Th2 immune pathway and ameliorated DSS-induced colitis. However, more studies should be conducted to validate the expression of other Th1- and Th2-related cytokines to identify the exact mechanism of action.

## 3. Discussion

In animals, DSS-induced colitis is used extensively as a reference procedure to study colitis. DSS, a polysaccharide comprising of a negative charge and sulfate moiety, has a molecular weight ranging from 5–1400 kDa. It was reported that the most severe form of colitis that closely resembles human UC can result when treating water and 40–50 kDa DSS [19,20]. This model is therefore very frequently used owing to its austerity, consistency, swiftness, and tractability. Hence, we employed this inflammation model in our study to explore the influence of Rg3-enriched red ginseng extract and PT regarding colitis.

PT, a flowering plant belonging to the buckwheat family, is most commonly termed “Chinese indigo” or “Japanese indigo”. It is called indigo because it is commonly used as a dye that gives blue/indigo color in fabrics. In addition to its dying properties, its stems and leaves possess antiphlogistic, antipyretic, anti-inflammatory, and depurative properties. PT has beneficial effects especially in inflammation where it reduces the permeability of capillaries and elevates the phagocytosis rate of white blood cells. PT is also used for skin disorders such as pimples and freckles, and disease conditions such as mumps, erysipelas, infantile convulsions and choreoathetosis (ICCA) syndrome, and febrile condition in children [21,22,23].

Ginseng is the most widely consumed herb in Southeast Asian countries, owing to its outstanding health-promoting qualities. Rg3-RGE, also a commonly consumed extract in the Korean peninsula, has been extensively examined because of its positive influences on health [24]. Since much literature has presented the effect of different types of combination extracts on colitis, working in an additive or synergistic manner [11,25], we sought to combine PT and Rg3-RGE to verify their effects when combined over the DSS-induced colitis sample (model).

In response to noxious stimuli such as foreign invaders or inflammation-inducing agents such as LPS or DSS, the interaction of LPS with its toll-like receptor (TLR4) triggers a sequence of inflammatory factors (*iNOS* and *COX-2*) and cytokines (*IL-1β*, *IL-6* and *TNF-α*) [26]. Previous literature has shown that the intensity of DSS introduced colitis in *iNOS* knockout mice or by treatment with specific *iNOS* inhibitors in mice was considerably inhibited [27]. In addition, immune-system hyper-activation terminates the release of pro-inflammatory cytokines such as *IL-1β, IL-6, IL-5, IL-13*, and *TNF-α*. The production of these chemicals requires timely management. Otherwise, they become major players in the erosion of intestinal epithelium [28]. As shown from our results, a combination of Rg3-RGE and PT synergistically hindered the interpretation of such pro-inflammatory mediators and cytokines in DSS-induced colitis mice’s macrophage cells, colon tissue, and plasma (Figure 2, Figure 3, Figure 5 and Figure 6).

We have previously reported the anti-inflammatory effects of Rg3-RGE [29], where mice induced with sepsis have been rescued. In that study, we have shown that Rg3-RGE has potently inhibited pro-inflammatory mediators and cytokines, and suppressed proteins in the *NF-κB* and *MAPK* pathway. In another study, we have reported that the anti-inflammatory effects of Rg3-RGE is mediated by the *RXRα-PPARγ* heterodimer [30]. Due to its potent anti-inflammatory effects, we sought to investigate whether its benefits can be further elevated by a combination with other herbs. PT has been reported for its antioxidant and anti-inflammatory activities [13]. Our findings in Figure 1B and Table 1 revealed that the PT contains kaempferol and quercetin, where both compounds were previously reported for its beneficial properties. Kaempferol is antitumor, antioxidant, and anti-inflammatory [31]. Quercetin has also been reported for its anti-inflammatory properties [32]. Hence, the ginsenoside Rg3 in Rg3-RGE, quercetin, and kaempferol and PT are known bioactive compounds in this mixture. Thus, our study suggests the synergism among these bioactive constituents, whose mechanism of action should be investigated in the future.

The primary pathway for inflammation elucidation is the conventional *NF-κB* pathway and the *MAPK*. Numerous studies have shown that suppression in these pathways is responsible for reduction with DSS-induced colitic mice [33,34]. Consistent with previous literature, our results have also shown that treatment with the mixture and single extracts inhibited the expression of *NF-κB* in both macrophage cells and colon tissue, and *MAPK* in macrophage cells (Figure 4 and Figure 7C).

*NLRP3*, the NOD-like receptor (*NLR*) family’s best-characterized member, interacts with an adaptor protein (apoptosis-associated speck as protein (*ASC*)). If triggered by noxious stimuli, *NLRP3*, *ASC*, and *pro-caspase 1* homogenize into a single complex called the inflammatory *NLRP3* which in turn cleaves *pro-caspase 1*; this cleavage causes *IL-1β* and *IL-18* to mature [35]. Once secreted, both *IL-1β* and *IL-18* are known sources to severe inflammation. *NLRP3* has been extensively studied due to its involvement in intestinal disorders such as colitis [6,36]. Our results have also shown that the single extracts and mixture treatment suppressed the levels of *NLRP3* both at the transcriptional and translational levels (Figure 7B,C).

Histologically, IBD is characterized by distortion of the cryptic epithelium, swelling, and thickening of mucosa and submucosa membranes and the excessive propagation of acute and chronic inflammatory cells (in mucosa and submucosa membranes) such as goblet cells of the intestinal lining [37]. Interestingly, the mixture of Rg3-RGE and PT remarkably recovered the intestinal damage caused by DSS treatment, or rather protected the colon from this disease. Colon shortening, which is also considered a hallmark for DSS-induced colitis in animal models, was recovered synergistically when mice were treated with the mixture (Figure 5).

Immune response initiation occurs when T-helper cells (CD4^+^) or Cytotoxic T lymphocyte (CTL; CD8^+^) recognize an antigen, most of which is of bacterial origin [38]. Neutrophils, macrophages, and CD8^+^ cells may be activated by the immune system’s failure to regulate T-cell responses. CD4^+^ and CD8^+^ cells divide physiologically into subpopulations of T-cells of Th1 and Th2; subtypes of Th1 cells are responsible for the secretion of *IL-1*, *IL-2*, *IL-12*, *IL-18*, *TNF-α*, and *IFNγ*. The elevation of these cytokines is mainly responsible for the clinical manifestations of UC and CD. However, bias against the release of Th2 cell cytokines (i.e., *IL-4*, *IL-5*, and *IL-10*) has been observed in UC. In addition, some studies have shown that Th2 cytokine release provides protection in mice with DSS-induced colitis [39,40]. Therefore, considering previous literature, our results have first shown the enhanced number of CD4^+^ cells and Tregs in splenocytes of mice (with DSS-induced colitis) (Figure 8A); second, Th1 cytokine (i.e., *IFNγ*) was found to be increased only in the DSS group and its levels decreased by Rg3-RGE + PT treatment (Figure 9A), whereas, Th2 cytokine, (i.e., *IL-4*) was remarkably enhanced by treatment with the mixture (Figure 9B). Our study and previous studies therefore support that a balance exists between Th1/Th2 cell cytokines in DSS introduced colitic mice. From the results obtained, it can be hypothesized as Rg3-RGE + PT protects mice from DSS-induced colitis by the *IL-4* protective mechanism.

## 4. Materials and Methods

### 4.1. Chemicals and Reagents

Dulbecco’s Modified Eagle Medium (DMEM), Fetal bovine serum (FBS), streptomycin and penicillin were obtained from Welgene (Daegu, Republic of Korea). Primers shown in Table 1 were acquired from Bioneer (Daejeon, Republic of Korea). Lipopolysaccharide (LPS), 3-(4,5-dimethylthiazol-2-yl)-2,5-diphenyltetrazoliumbromide (MTT), and dextran sulfate sodium (DSS) were from Sigma-Aldrich (Catalogue# 42867, Molecular weight is 40,000 Da, St. Louis, MA, USA). Antibodies for Western blot, *iNOS* (#2982), *COX-2* (#4842), *p-TAK1* (#4531), *p-NFκB p65* (#3033), *IRAK-1* (#4359), *p-IKKα/β* (#2697), *p-IκBα* (#2859), *β-actin* (#4967), *T-ERK* (#9102), *p-ERK* (#9101), *T-JNK* (#9252), *p-JNK* (#9251), *T-p38* (#9212), *p-p38* (#9211), *NLRP3* (#15101), HRP-linked antibody (#7074) were all obtained from Cell Signaling Technology (Danvers, MA, USA). Antibodies conjugated with fluorescence for flow cytometry analysis, PE-Cy5 anti-CD3 (145-2C11; 553065), PE anti-CD4 (RM4-5; 553049), FITC anti-CD8 (53-6.7; 553031), FITC anti-CD25 (3C7; 558689), FITC anti-CD19 (ID3; 553785) and PerCP-Cy5.5 anti-CD69 (H1.2F3; 561931), and CD3e monoclonal antibody (145-2C11; 553058) were obtained from BD Biosciences (San Diego, CA, USA).

### 4.2. Sample Preparation

Rg3-enriched red ginseng extract was prepared and analyzed via HPLC according to our previous study [38] as shown in Figure 1A. PT was prepared by boiling the leaves of the plant with 70% Ethanol and thereafter a concentrated extract was prepared by condensation and later a powdered form was obtained by lyophilization. Thereafter, PT extract was analyzed via UPLC–QTofMS for its active components as shown in Figure 1B. Analysis was carried out using a UPLC system (Waters Corp; Milford, MA, USA). The analysis was conducted by injecting sample into an AQUITY UPLC^®^ BEH C_18_ (2.1 × 100 mm × 1.7 μm) column. The mobile phase consisted of distilled water as solvent A, and acetonitrile as solvent B. The column was eluted with a gradient elution at a flow rate of 0.4 mL/min as stated: initial, solvent A 90%; solvent B 10%, 0–1 min, solvent A 90%; solvent B 10%, 1–11 min, solvent A 10%; solvent B 90%, 11–11:30 min, solvent A 0%; solvent B 100%, 11:30–13:40 min, solvent A 0%; solvent B 100%, 13:40–13:50 min, solvent A 90%; solvent B 10%, 13:50–15:00 min solvent A 90%; solvent B 10%.

### 4.3. Cell Culture

The murine macrophage cell line, RAW 264.7, from American Type Culture Collection (ATCC, Manassas, VA, USA) was cultured in DMEM supplemented with 8% FBS, 100 IU/mL penicillin and 100 μg/mL streptomycin sulfate; the cells were incubated in a humidified 5% CO_2_ atmosphere at 37 °C.

### 4.4. Nitric Oxide (NO) Assay

Nitric oxide was measured by the method for the Griess reaction assay. RAW 264.7 cells were seeded in a 96-well plate and incubated with or without LPS (0.1 μg/mL) in the absence or presence of Rg3-RGE + PT at indicated concentrations for 18 h. Remaining steps were done as reported in our previous study [41]. Briefly, supernatant was collected and added with Griess reagent for the detection of NO. Absorbance of the resulting solution was checked at an absorbance of 540 nm using a microplate reader (Versamax; Molecular Devices, San Jose, CA, USA).

### 4.5. Cell Viability (MTT) Assay

To determine the cytotoxic effects of samples, cell viability assay was performed using MTT reagent which was added to the culture medium at a final concentration of 0.1 mg/mL. After 4 h of incubation at 37 °C in 5% CO_2_, the resulting violet-colored crystals were dissolved in dimethyl sulfoxide (DMSO), 100 μL/well and the absorbance values measured at 560 nm using a microplate reader (Versamax; Molecular Devices, San Jose, CA, USA).

### 4.6. DSS-Induced Colitis Model and Ginseng Mixture Treatment Regimens

Male C57BL/6 mice (6–8 weeks old; 18–20 g) were purchased from Charles River (Orient Biotechnology, Gyeonggi-do, South Korea) and housed in a specific-pathogen-free barrier facility at 21 ± 2 °C with a relative humidity of 60 ± 10% under a 12h light and dark cycle. Feed and water were provided *ad libitum*. All animal care and experimental procedures were performed in accordance with internationally accepted guidelines on the use of laboratory animals (IACUC) and the protocols were approved by the Animal Care Committee of the College of Veterinary Medicine, Kyungpook National University, Daegu, Republic of Korea (January 18, 2018: approval number: KNU2018-002). For the experiment, mice were divided into 6 groups (*n* = 6 in each group) as follows: (1) control; (2) 3% DSS; (3) positive control, sulfasalazine (75 mg/kg) + DSS; (4) Rg3-RGE (20 mg/kg) + DSS; (5) PT (300 mg/kg) + DSS; and (6) mixture of Rg3-RGE + PT + DSS. All groups excluding group 1 were administered 3% DSS in drinking water *ad libitum* for 7 days after a period of acclimatization. Oral administration of sulfasalazine, Rg3-RGE, PT, and the mixture was initiated simultaneously on the same day as DSS administration. On day 7th of experiment, all mice were euthanized, and blood, colon, and spleen tissues collected for further experimentations.

### 4.7. RNA Extraction and Polymerase Chain Reaction (PCR)

RAW 264.7 cells were pretreated with or without Rg3-RGE + PT at the indicated concentrations for 30 min and then stimulated with LPS (0.1 μg/mL) for 18 h. Later steps were according to our previous study [37]. Briefly, total RNA was extracted and reverse transcribed with a kit to obtain the cDNA (Bioneer, Daejeon, Republic of Korea). Resultant cDNA undergoes polymerase chain reaction (PCR), and the PCR product was run on 1% ethidium bromide stained-agarose gel. Gel images were developed (General Electrics, Boston, MA, USA). Images were quantified with Image J version 1.53a (NIH, Montgomery County, MD, USA) and relatively compared with GAPDH, which was used as the housekeeping gene. Experiments were repeated three times. Colon of mice were subjected to qRT-PCR using Bio-Rad Real-Time Thermal Cycler CFX96 (Bio-Rad, Hercules, CA, USA). The PCR primer sequences are given in Table 2.

### 4.8. Western Blot Analysis

RAW 264.7 cells were treated with Rg3-RGE + PT or left untreated in the presence or absence of LPS (0.1 μg/mL). Later steps were according to our previously reported study [30]. Briefly, protein was extracted from cells and separated via 10% SDS-PAGE. Proteins were transferred to PVDF membranes, blocked, and incubated with the respective primary antibodies (1:1000). The next day, membranes were washed and incubated with secondary antibody (1:3000) for 75 min before developing in a gel developer (General Electrics, Boston, MA, USA).

### 4.9. Gross Examination of Colon and Disease Activity Index (DAI)

For the gross examination of colon, mice were sacrificed, and the colon was removed. The length was then measured and plotted on a graph. Weight measurement was performed every day for 7 days and mice were macroscopically observed for their general disposition, consistency of stool and presence of blood in stools (hematochezia). Thereafter, DAI, a combined score of weight loss, stool consistency, and presence of blood in stools, was determined according to Sann et al. [42].

### 4.10. Enzyme Linked Immunosorbent Assay (ELISA) for Cytokines

The levels of *IL-5*, *IL-13*, *IL-1β* and *TNF-α* in the plasma of DSS-induced colitis mice were measured using Mouse *IL-5*, Mouse *IL-13*, Mouse *IL-1β* and Mouse *TNF-α* Quantikine ELISA kits (R&D Systems, Minneapolis, MN, USA), respectively, according to the manufacturers protocols. All samples were analyzed in triplicates and normalized to the total protein content of the sample, expressed as pg/mg protein.

### 4.11. Hematoxylin and Eosin (H&E) Staining

After euthanasia, colon tissues were harvested in 10% Neutral buffered formalin and the tissues processed for basic H&E staining, using established protocols [43].

### 4.12. Fluorescent Antibody Cell Sorting (FACS)

The spleens from mice were lightly pulverized using a BD syringe plunger and passed through a cell strainer (pore size, 70 µm) into DPBS (Dulbecco’s phosphate buffered saline) and centrifuged at 245 g for 5 min at 20–25 °C. This experiment was further carried out according to our previously reported protocol [37]. Briefly, red blood cells were lysed and centrifuged to obtain splenocytes. Splenocytes were adjusted to 1 × 10^5^ cells and stained with specific antibodies, and resuspended in FACS buffer (2% FBS in phosphate buffered saline). Lymphocytes were gated and analyzed with BD FACSAriaIII^TM^ (BD Biosciences, San Jose, CA, USA). Unstained cells were used as a control for the negative population.

### 4.13. Estimation of Th1/Th2 Cytokines Using ELISA

Th1/Th2 cytokines (i.e., *IL-4* and *IFN-γ*) were measured using ELISA. 96 well plates were coated with 0.5 µg/mL CD3e mAb before incubation with splenocytes overnight. The MoAbs-based mouse interleukin ELISA kits (R&D system, Minneapolis, MN, USA) was used the following day to determine their levels, according to manufacturer’s instructions.

### 4.14. Statistical Analysis

Data are presented as mean ± SD. One-way ANOVA (Analysis of variance) and Dunnett’s test were applied to statistically evaluate the data. Statistical analyses with *** *p* < 0.001 and ** *p* < 0.05 and * *p* < 0.01 were considered significant compared to LPS and DSS group; ^#^
*p* < 0.01 when compared to Rg3-RGE or PT alone.

## 5. Conclusions

To conclude, this is the first research to display the effects of Rg3-RGE, PT, and the combined mixture on DSS-induced colitis in mice, to our best knowledge. Synergistically, Rg3-RGE + PT inhibited expression of NO, *iNOS*, *COX-2*, *IL-1β*, *IL-6*, *IL-5*, *IL-13*, and *TNF-α* in the DSS-induced colitis mice’s macrophage cells, plasma, and colon tissue. In addition, these attenuating effects were acting through the *NF-κB* and *MAPK*, and suppressed by *NLRP3*. The Rg3-RGE + PT group experienced synergistic recovery of the macroscopic parameters (DAI, length of the colon, and weight loss) and colon microscopic lesions. Rg3-RGE + PT also up-regulated mice’s CD4^+^ cells, and the Th2 cytokine, *IL-4*, was identified as the target to protect mice from DSS-induced colitis.

## Figures and Tables

**Figure 1 molecules-25-05230-f001:**
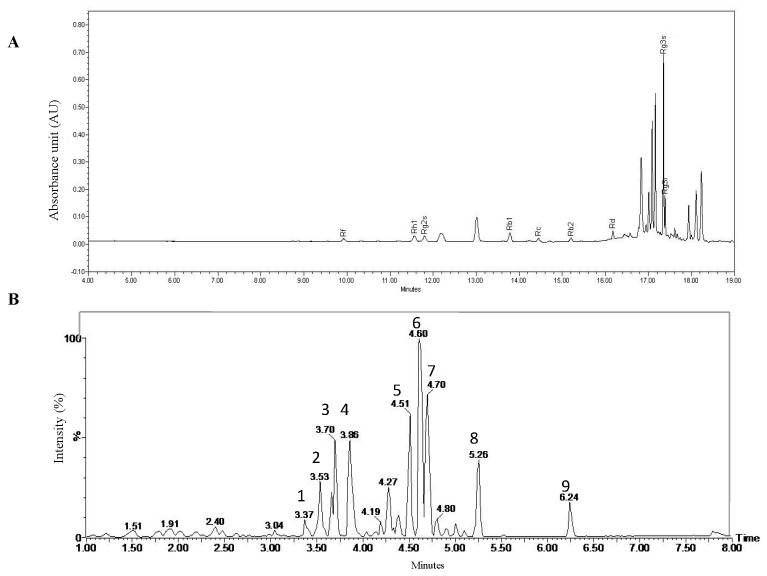
Bioactive compound analysis in Rg3-RGE and *Persicaria tinctoria* (PT). Chromatogram for the HPLC analysis of Rg3-RGE (**A**) with the respective ginsenosides identified as labeled on their respective peaks. UPLC–QTofMS analysis of PT and their identified peaks (**B**). Constituents of PT are summarized in Table 1.

**Figure 2 molecules-25-05230-f002:**
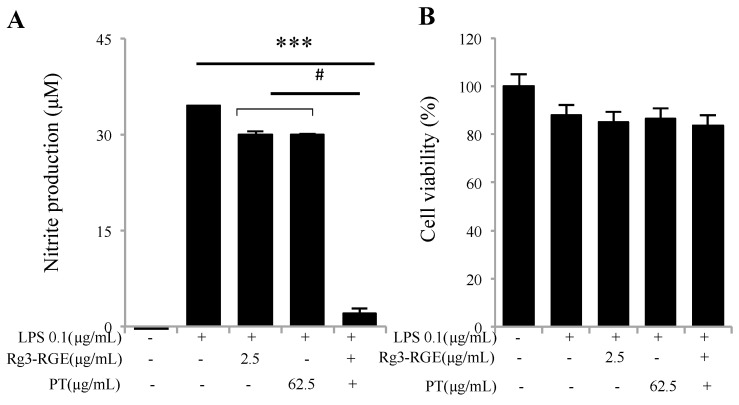
Suppression of nitric oxide (NO) by Rg3-RGE + PT. RAW 264.7 cells were pre-incubated with Rg3-RGE + PT for 30 min and then stimulated with LPS for 18 h. Cell supernatant was then mixed with equal amounts of Griess reagent and the NO production measured (**A**). Effects of Rg3-RGE + PT on cell viability were measured by MTT (3-(4,5-dimethylthiazol-2-yl)-2,5-diphenyltetrazolium bromide) assay (**B**). *** *p* < 0.001 is considered significant compared to LPS and ^#^
*p* < 0.01 compared to inhibition by only Rg3-RGE or PT. Values in bar graphs represent mean ± SD of three independent experiments. Triplicates were completed within 10 days.

**Figure 3 molecules-25-05230-f003:**
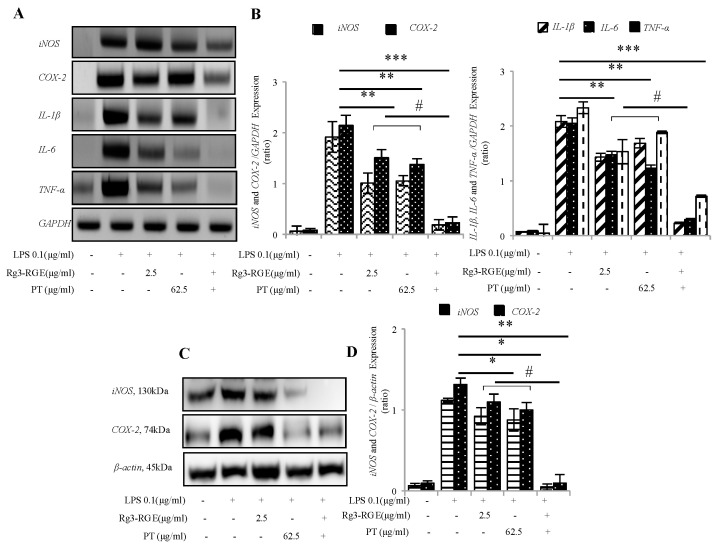
Rg3-RGE + PT hindered the expression of pro-inflammatory mediators and cytokines. RAW 264.7 cells were pre-incubated with Rg3-RGE + PT for 30 min and then stimulated with LPS for 18 h. Total RNA and proteins from cells were extracted and the mRNA and protein expression of *iNOS*, *COX-2* and *IL-1β*, *IL-6* and *TNF-α* determined by (**A**,**B**) RT-Polymerase Chain Reaction (PCR), and (**C**,**D**) Western blot. *GAPDH* (Glyceraldehyde 3-phosphate dehydrogenase) and *β-actin* were used as internal control. Images are representative of three independent experiments. Values in bar graphs represent mean ± SD of three independent experiments. *** *p* < 0.001, ** *p* < 0.05 and * *p* < 0.01 considered significant compared to LPS and ^#^
*p* < 0.01 compared to inhibition by only Rg3-RGE or PT. Triplicates for RT-PCR were completed within 10 days whereas Western blot analysis were completed within 3 weeks.

**Figure 4 molecules-25-05230-f004:**
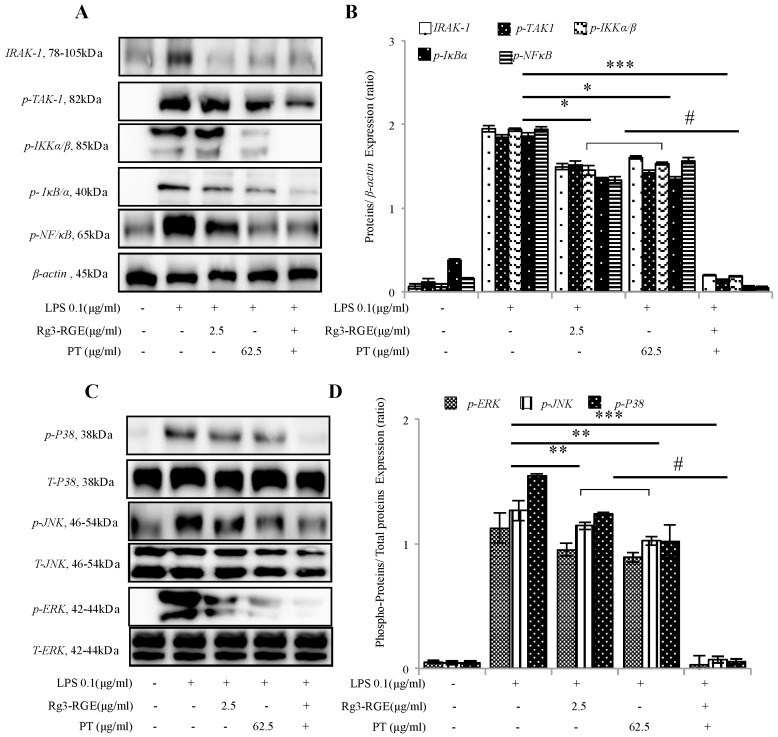
Signal transduction of Rg3-RGE + PT by the *NF-κB* and *MAPK* pathways. RAW 264.7 cells were pre-incubated with Rg3-RGE + PT extract for 30 min and then stimulated with LPS for 18 h. Nuclear and cytosolic proteins from cells were extracted by NE-PER^®^ nuclear and cytosolic extraction reagents. *β-actin* was used as an internal control. Images are representative of three independent experiments. (**A**,**B**) *NF-κB* pathway, and (**C**,**D**) *MAPK* pathway. Values in bar graphs represent mean ± SD of three independent experiments. *** *p* < 0.001, ** *p* < 0.05 and * *p* < 0.01 is considered significant compared to LPS and ^#^
*p* < 0.01 compared to inhibition by only Rg3-RGE or PT. Triplicates for Western blot analysis were completed within 3 weeks.

**Figure 5 molecules-25-05230-f005:**
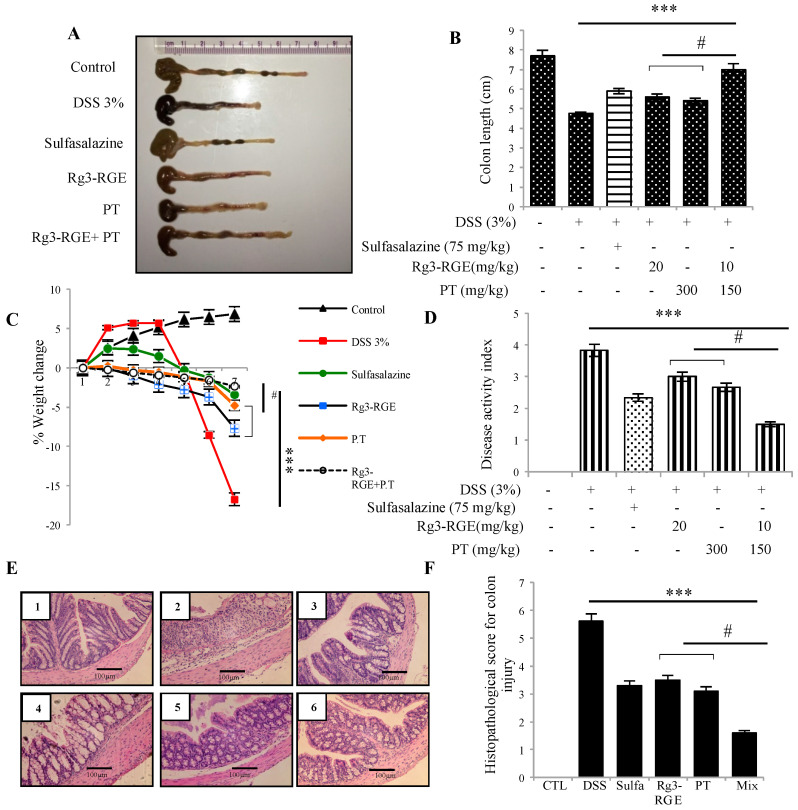
Improvement of colon length, disease activity index (DAI) and histopathological lesions by Rg3-RGE + PT treatment. C57BL/6 mice were given 3% DSS in drinking water for 7 days along with p.o. treatment of the extracts. After 7 days, mice were euthanized, and the colon tissue were extracted for macroscopic and microscopic lesion assessment. (**A**) Gross examination of colon length (**B**) Quantification of colon length. (**C**) % Change in the body weight of mice. (**D**) DAI. (**E**) Hematoxylin and Eosin (H&E) staining of colon tissues (1) Control, (2) DSS, (3) Sulfasalazine + DSS, (4) Rg3-RGE + DSS, (5) PT + DSS and (6) Rg3-RGE + PT + DSS. Scale bar = 100 µm. (**F**) Histopathological score for colon damage. *** *p* < 0.001 considered significant compared to DSS group and ^#^
*p* < 0.01 compared to inhibition by only Rg3-RGE or PT. Macroscopic analysis were conducted the same day of termination.

**Figure 6 molecules-25-05230-f006:**
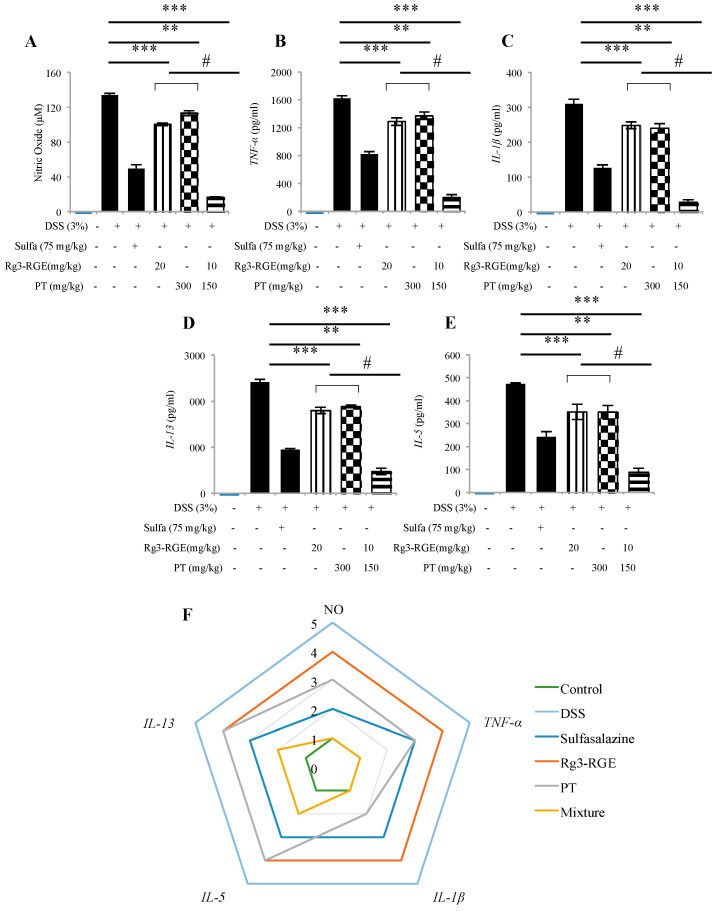
Rg3-RGE + PT abolished the expression of pro-inflammatory mediators and cytokines in plasma of DSS mice. Plasma from blood was extracted and the expression of NO, *TNF-α*, *IL-1β*, *IL-13* and *IL-5* were determined by Enzyme Linked Immunosorbent Assay (ELISA) and colorimetric assay (**A**–**E**) where (**A**) NO expression, (**B**) *TNF-α* expression, (**C**) *IL-1β* expression, (**D**) *IL-13* expression, and (**E**) *IL-5* expression. Radar chart for normalized data trend fitting analysis of the serum biomarkers (**F**). Values in bar graphs represent mean ± SD of three independent experiments. *** *p* < 0.001 and ** *p* < 0.05 were considered significant compared to DSS group and ^#^
*p* < 0.01 compared to inhibition by only Rg3-RGE or PT. ELISA were conducted within 48 h of termination.

**Figure 7 molecules-25-05230-f007:**
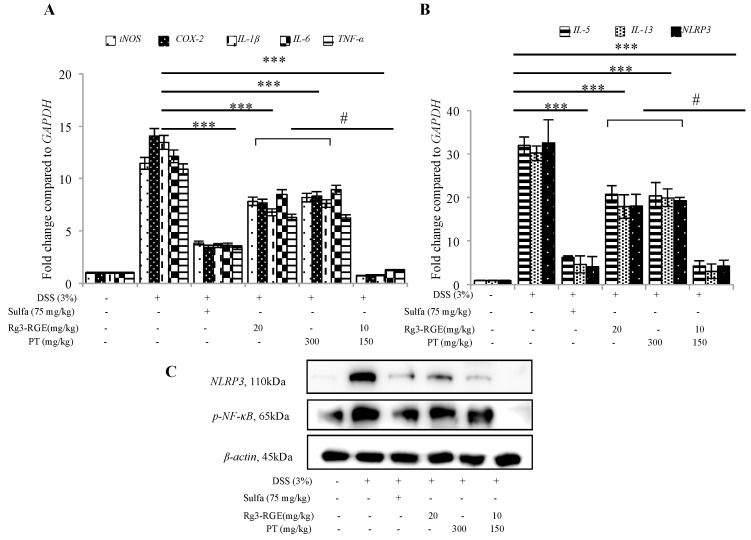
Inhibition of mRNA of pro-inflammatory cytokines, *NLRP3* and *NF-κB* protein expression by Rg3-RGE + PT. Total RNA and proteins were extracted from colon tissue of DSS-treated mice. mRNA was subjected to qRT-PCR and the expression of pro-inflammatory mediators and cytokines determined by SYBR (Synergy Brand, Inc) green fluorescence. Nuclear and cytosolic proteins were extracted using NE-PER^®^ Nuclear and cytosolic extraction reagents. (**A**,**B**) qRT-PCR for *iNOS*, *COX-2*, *IL-1β*, *IL-6*, *TNF-α*, *IL-5, IL-13*, and *NLRP3*. (**C**) Western blot for *NLRP3* and *NF-κB*. *β-actin* was used as an internal control. Images are representative of three independent experiments. Values in bar graphs represent mean ± SD of three independent experiments. *** *p* < 0.001 was considered significant compared to DSS group and ^#^
*p* < 0.01 compared to inhibition by only Rg3-RGE or PT. Experiments were completed within one week of termination.

**Figure 8 molecules-25-05230-f008:**
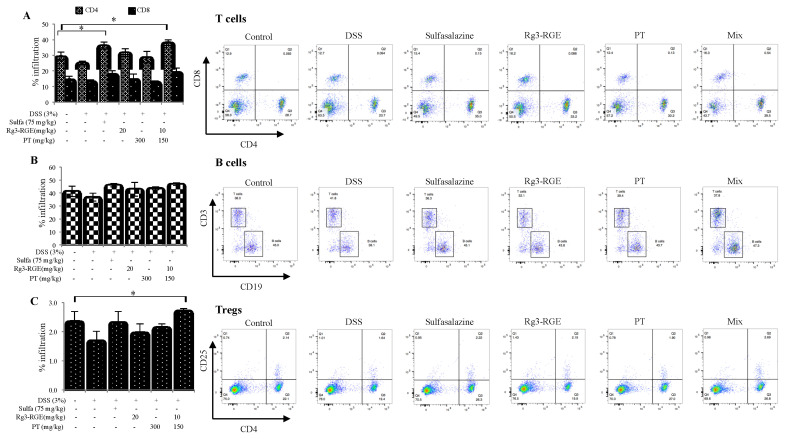
Effects of Rg3-RGE + PT on immune cell subtypes in the spleen. (**A**) Percentage of CD4^+^ and CD8^+^ cells, (**B**) CD19^+^ cells and (**C**) CD4^+^CD25^+^ cells in the spleen. * indicates a significance of *p* < 0.01 for treatment group when compared to the DSS-treated group. Values in bar graphs represent mean ± SD of four independent experiments. Analysis was conducted on the same day of termination.

**Figure 9 molecules-25-05230-f009:**
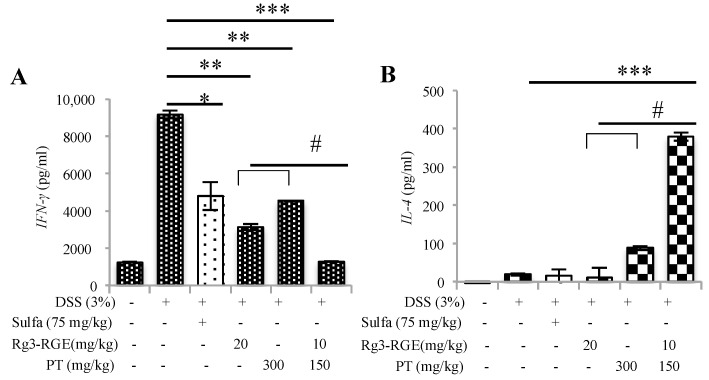
Effects of Rg3-RGE + PT on lymphokine secretion. (**A**) Splenocyte supernatant levels of *IFN-γ* in the Sulfasalazine, Rg3-RGE, PT and Rg3-RGE + PT groups as determined by ELISA. (**B**) Splenocyte supernatant levels of *IL-4* in the Rg3-RGE + PT group as determined by ELISA *** *p* < 0.001, ** *p* < 0.05 and * *p* < 0.01 when compared to the DSS-treated group and ^#^
*p* < 0.01 compared to the inhibition by only Rg3-RGE or PT. Values in bar graphs represent mean ± SD of four independent experiments. Experiments were conducted within 72 h of termination.

**Table 1 molecules-25-05230-t001:** Constituents of PT analyzed via UPLC–QTofMS analysis.

Peak No.	ESI-MS *R*_T_ (min)	UV (nm)	Detected Ion (*m*/*z*) [M−H]^−^	Calculated Ion (*m*/*z*) [M−H]^−^	Fragments	Molecular Formula	Identification
1	3.37	254, 351	477.0648	477.0669	301	C_21_H_18_O_13_	Quercetin-3-*O*-β-d-glucuronide
2	3.53	254, 354	607.1312	607.1299	505, 463, 301	C_27_H_28_O_16_	Quercertin-3-*O*-[6′′-*O*-(3-hydroxy-3-methylglutaryl)-β-d-glucopyranoside]
3	3.70	268, 338	477.1031	477.1033	315, 299	C_22_H_22_O_12_	Isorhamnetin-3-*O*-β-d-glucopyranoside
4	3.86	265, 346	591.1360	591.1350	447, 285	C_27_H_28_O_15_	Kaempferol-3-*O*-[6′′-(3-hydroxy-3-methylglutaryl)-β-d-glucopyranoside]
5	4.51	277, 339	475.1005	475.0877	313	C_22_H_20_O_12_	3,5,4′-Trihydroxy-6,7-methylenedioxyflavone-3-*O*-β-d-glucopyranoside
6	4.60	277, 340	619.1486	619.1299	475, 313	C_28_H_28_O_16_	3,5,4′-Trihydroxy-6,7-methylenedioxyflavone-3-*O*-[6′′-(3-hydroxy-3-methylglutaryl)- β-d-glucopyranoside]
7	4.70	277, 339	517.1122	517.0982	475, 313	C_24_H_22_O_13_	3,5,4′-Trihydroxy-6,7-methylenedioxyflavone-3-*O*-[6′′-(acetyl)-β-d-glucopyranoside]
8	5.26	274, 339	531.0790	531.0775	471, 313	C_24_H_20_O_14_	3,5,4′-Trihydroxy-6,7-methylenedioxyflavone-3-*O*-[2′′-(acetyl)-β-d-glucuronide]
9	6.24	274, 351	313.0327	313.0348	no fragment	C_21_H_18_O_13_	3,5,4′-Trihydroxy-6,7-methylenedioxyflavone

**Table 2 molecules-25-05230-t002:** Primer sequences used in this study.

Gene	Primer	Oligonucleotide Sequence (5′-3′)	Accession Number
*GAPDH*	Forward	5′-CAATGAATACGGCTACAGCAAC-3′	BC023196.2
	Reverse	5′-AGGGAGATGCTCAGTGTTGG-3′	
*iNOS*	Forward	5′-CCCTTCCGAAGTTTCTGGCAGCAGC-3′	NM_001313922.1
	Reverse	5′-GGCTGTCAGAGCCTCGTGGCTTTGG-3′	
*COX-2*	Forward	5′- CAAGACGCCACATCCCCTAT -3′	LC061973.1
	Reverse	5′- ATTTAGTCGGCCTGGGATGG -3′	
*IL-1β*	Forward	5′-CAGGGTGGGTGTGCCGTCTTTC-3′	NM_008361.4
	Reverse	5′-TGCTTCCAAACCTTTGACCTGGGC-3′	
*TNF-α*	Forward	5′-TTGACCTCAGCGCTGAGTTG-3′	NM_001278601.1
	Reverse	5′-CCTGTAGCCCACGTCGTAGC-3′	
*IL-6*	Forward	5′-GTACTCCAGAAGACCAGAGG-3′	NM_001314054.1
	Reverse	5′-TGCTGGTGACAACCACGGCC-3′	
*IL-5*	Forward	5′-GAAGTGTGGCGAGGAGAGAC-3′	BC125366.1
	Reverse	5′- GCACAGTTTTGTGGGGTTTT-3′	
*IL-13*	Forward	5′-AGCATGGTATGGAGTGTGGA-3′	NM_008355.3
	Reverse	5′-TTGCAATTGGAGATGTTGGT-3′	
*NLRP3*	Forward	5′-TGCTCTTCACTGCTATCAAGCCCT-3′	NM_145827.4
	Reverse	5′-ACAAGCCTTTGCTCCAGACCCTAT-3′	

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
