# Peer review of "Alleviation of Ulcerative Colitis Potentially through th1/th2 Cytokine Balance by a Mixture of Rg3-enriched Korean Red Ginseng Extract and Persicaria tinctoria"

_molecules, 2020, doi:10.3390/molecules25225230_

Round 1

Reviewer 1 Report

This study has aimed at showing the anti-inflammatory potential of the traditional medicine Korean red ginseng extract and Persicaria tinctoria in ulcerative colitis. While the topic is important, the manuscript contains various concerns. Please, find detailed comments below.

  • Indigo is a well-known sensitizer, which is common also to other plants. Please, consider whether the Th2-type response could be due to hypersensitivity. The shift away from pro-inflammatory response is naturally beneficial from the view of colitis but may be harmful in other ways. These risks should be carefully considered.

  • Please, clarify the concurrent analysis of HPLC peaks as bioactive.

  • The significance of mRNA level data should be appropriately interpreted. All cytokines may never become translated into proteins.

  • Molecular weights of western blot bands should be indicated in Figures and bands should be quantitated and normalized against their internal controls.

  • The number of parallel samples as well as individual experiments (performed days, weeks or months apart from each other) should be clearly reported in Figure legends. According to the number of samples per group as well as the data normality, the most optimal test must be selected and a statistician should be consulted in unclear situations. Probably, a parametric test is not suitable for these data but a non-parametric test, such as the Mann-Whitney U-test, should be preferred. It requires at least four samples per group and if experiments were performed in triplicates, the number of samples per group can be increased by combining data from different experiments, and inter-experiment level variation can be reduced by normalizing results of each experiments against to its control.

  • Symbols in Fig. 5C should differ more from each other.

  • Groups to be compared could be connected e.g. by horizontal lines above the bars.

  • Antibodies should be specified e.g. by catalog numbers and/or clones.

  • Centrifugation should be reported as g forces instead of rpms that are dependent on the rotor diameter.

  • The manuscript would benefit from a careful language revision.

Author Response

Respected reviewers,

We are humbly obliged to you for taking your precious time out for reviewing our manuscript and giving us very helpful revisions that will ascend the quality of our manuscript and research.

Kindly please review our corrections to the comments that has been raised below.

Reviewer 1

This study has aimed at showing the anti-inflammatory potential of the traditional medicine Korean red ginseng extract and Persicaria tinctoria in ulcerative colitis. While the topic is important, the manuscript contains various concerns. Please, find detailed comments below.

  1. Indigo is a well-known sensitizer, which is common also to other plants. Please, consider whether the Th2-type response could be due to hypersensitivity. The shift away from pro-inflammatory response is naturally beneficial from the view of colitis but may be harmful in other ways. These risks should be carefully considered.

Answer: Respected Reviewer, we thank you for your comment. We agree with you completely. However, it is not the true indigo. It belongs to family of buckwheat while true indigo belongs to fabaceae family. Indigo carmine (sodium indigotindisulfonate), a blue dye from indigofera polygonacium (True indigo), is widely used by surgeons to identify and to examine the urinary tract and is considered biologically inert and extremely safe with minor hypersensitivity problems. According to our extensive internet searches, we have not found out any report on indigo obtained from P.T as a source of hypersensitivity please.

  1. Please, clarify the concurrent analysis of HPLC peaks as bioactive.

Answer: Dear reviewer, thank you for pointing this out, we have added the labels in Figure 1B, corresponding to the peak numbers as stated in Table 1. We thank you for your suggestion.

  1. The significance of mRNA level data should be appropriately interpreted. All cytokines may never become translated into proteins.

Answer: Respected Reviewer, We thank you for your comment. We agree with you completely. Cyclooxygenase (COX) is the key enzyme in the synthesis of prostaglandins (PGs). COX-2 is an inducible enzyme expressed in macrophages, fibroblasts, and other cell types in inflammation. COX-2 expression is induced by phorbol esters and proinflammatory cytokines, including interleukin (IL)-1β and tumor necrosis factor (TNF)-a. Both NO and prostaglandins (PGs) are well known to be important mediators of acute and chronic inflammation and are synthesized by nitric oxide synthase (NOS) and cyclooxygenase (COX) enzymes, respectively. iNOS and COX-2 are upregulated in response to inflammatory and pro-inflammatory mediators and when translated into proteins are activators of various inflammatory pathways. That is why we have only shown the translated forms of COX-2 and iNOS please.

  1. Molecular weights of western blot bands should be indicated in Figures and normalized with their internal control.

Answer: Respected Reviewer, We thank you for your comment. We have added the molecular weights for western blot bands in Figure 3, 4 and 7 please. β-actin was taken as our internal control please.

  1. The number of parallel samples as well as individual experiments (performed days, weeks or months apart from each other) should be clearly reported in Figure legends. According to the number of samples per group as well as the data normality, the most optimal test must be selected and a statistician should be consulted in unclear situations. Probably, a parametric test is not suitable for these data but a non-parametric test, such as the Mann-Whitney U-test, should be preferred. It requires at least four samples per group and if experiments were performed in triplicates, the number of samples per group can be increased by combining data from different experiments, and inter-experiment level variation can be reduced by normalizing results of each experiments against to its control.

Answer: Respected Reviewer, we thank you for your comment and we highly appreciate your suggestion regarding the statistics. We have added statements in the figure legends indicating the period of time when the experiments have been performed, as shown in red. We have added an extra analysis on the inter-experiment level variation in figure 6F as it is considered one of the more important indicator in our in vivo studies. This indeed provides a better comparison among groups and among the cytokine levels. From our findings the mixture of both Rg3-RGE and PT have stronger effects as compared to their treatment alone based on the scoring especially in NO, TNF-α and IL-1β. We thank you again for your suggestion. We have made amendments in the results section and figure legend, as shown in red.

  1. Symbols in Fig. 5C should differ more from each other.

Answer: Respected Reviewer, we thank you for your comment. We have improved figure 5C visually for easy identification of groups.

  1. Groups to be compared could be connected e.g. by horizontal lines above the bars.
  2.  

Answer: Respected Reviewer, We thank you for your comment. We have put the horizontal line for the groups to be compared in all the figures.

  1. Antibodies should be specified e.g. by catalog numbers and/or clones.

Answer: Dear reviewer, we thank you for your suggestion. We have added the catalog numbers for the antibodies used for western blot. Antibodies used for flow cytometry are also stated with their respective clones and catalog numbers in lines 305-313. We thank you for your suggestion.

  1. Centrifugation should be reported as g forces instead of rpms that are dependent on the rotor diameter.
  2.  

Answer: Dear reviewer, thank you for your suggestion. We have changed the rpm to g force in 396. We thank you for your precious suggestion.

  1. The manuscript would benefit from a careful language revision.

Answer: Respected Reviewer, We thank you for your useful concern over this query. Our manuscript has been edited for English by Editage Company please. We have attached the certificate below.

Respected reviewer, very humbly, we hope that we have answered to all the queries in a careful and detailed manner. We would again like pay a humble gratitude for reviewing our manuscript and giving useful comments for quality of manuscript.

With all due respect and regards,

Man Hee Rhee.

Reviewer 2 Report

The authors have studied the effects of red ginseng extract enriched with Rg3 (Rg3-RGE) and a  Persicaria tinctoria (PT) extract, alone or in combination, upon inflammation in RAW 264.7 cells in vitro and  Dextran Sodium Sulphate (DSS)-induced colitis in mice. Their findings that Rg3-RGE + PT acted in synergy to modulate the mitogen-activated protein kinase and nuclear factor κ B pathways and block inflammation in RAW 264.7 cells. Moreover, they suppressed, pro-inflammatory mediators/cytokines, and the NLRP3 inflammasome in DSS-treated mice ameliorated damage and disruption to the colon. They suggest that this mixture can be used for the prevention of UC as a prophylactic/ therapeutic supplement.

While this is a thorough and well-documented piece of work and the combination of Rg3-RGE and PT shows potential, the authors can only suggest that the mixture may have prophylactic properties because in the studies in vitro and in vivo the mixture was given in tandem with the pro-inflammatory factor rather than after onset of inflammation or tissue disruption.

The combination of Rg3-RGE and PT extract could prevent the development of inflammatory responses in vitro and in vivo. However, it is unclear what bioactive factors are responsible for this protection or how they may act in combination to give protection. This is critical given that PT is a mixture of components as is PTRg3-RGE to a lesser extent and needs to be discussed within the paper. Otherwise, it is a case of two ill-defined extracts combining to have a beneficial effect. How robust are the findings? Would they be reproducible if other preparations of PT were made and used?  Some hypotheses as to the responsible factors and mode of action need to be proposed.

Ln 68-68         How representative would this be if multiple batches prepared independently were compared? How much component variation is likely to occur? What is known about the bioactive properties of the identified compounds?

ln 148              Why were doses of Rg3-RGE and PT altered when given in combination?

                        Markers for figure 5C are practically indistinguishable from each other. These need to be changed.

Ln 158             ‘To investigate whether the in vitro results were due to colitis induction in mice’. What?

Ln 176-177     ‘the mixture brings on signal propagation via NF-κB pathway.’  The mixture blocks or inhibits signal propagation through the NF-κB pathway?

Ln 193-195     For clarity give appropriate figure reference for each set of data.

Ln 292             Give details of the DSS used in the study [product number, 40kDa?]. Were daily water and DSS intakes monitored for the test and control groups?

Author Response

Reviewer 2

The authors have studied the effects of red ginseng extract enriched with Rg3 (Rg3-RGE) and a Persicaria tinctoria (PT) extract, alone or in combination, upon inflammation in RAW 264.7 cells in vitro and Dextran Sodium Sulphate (DSS)-induced colitis in mice. Their findings that Rg3-RGE + PT acted in synergy to modulate the mitogen-activated protein kinase and nuclear factor κ B pathways and block inflammation in RAW 264.7 cells. Moreover, they suppressed, pro-inflammatory mediators/cytokines, and the NLRP3 inflammasome in DSS-treated mice ameliorated damage and disruption to the colon. They suggest that this mixture can be used for the prevention of UC as a prophylactic/ therapeutic supplement.

While this is a thorough and well-documented piece of work and the combination of Rg3-RGE and PT shows potential, the authors can only suggest that the mixture may have prophylactic properties because in the studies in vitro and in vivo the mixture was given in tandem with the pro-inflammatory factor rather than after onset of inflammation or tissue disruption.

  1. The combination of Rg3-RGE and PT extract could prevent the development of inflammatory responses in vitro and in vivo. However, it is unclear what bioactive factors are responsible for this protection or how they may act in combination to give protection. This is critical given that PT is a mixture of components as is PT+Rg3-RGE to a lesser extent and needs to be discussed within the paper. Otherwise, it is a case of two ill-defined extracts combining to have a beneficial effect. How robust are the findings? Would they be reproducible if other preparations of PT was made and used? Some hypotheses as to the responsible factors and mode of action need to be proposed. How representative would this be if multiple batches prepared independently were compared? How much component variation is likely to occur? What is known about the bioactive properties of the identified compounds?
  2.  

Answer: Respected Reviewer, We thank you for your useful concern over this query. We absolutely agree with your concerns that it is an Rg3-RGE+PT mixture. Please be assured that we have already reported in our previous studies about the anti-inflammatory effects of Rg3-RGE in sepsis model as follows:

In addition, the anti-inflammatory activities of Persicaria tinctoria is also being reported previously as follows;

The robust findings in this study is primarily the synergetic effects on two already reported anti-inflammatory extracts in combination.

We are positive that if these components are otherwise prepared, the findings can be reproducible.

In case of Persicaria Tinctoria, based on HPLC results, Kaempferol and quercetin are active components responsible for its anti-inflammatory activity. Moreover, a large number of literature is present on the anti-inflammatory effects of kaempferol and quercetin individually as follows:

In addition, we have already reported the anti-inflammatory activities of g3-RGE separately as discussed above.

Therefore, kamepferol and quercetin in Persicaria Tinctoria and Rg3 in Rg3-RGE are the known, reported active components in the mixture for anti-inflammation.

If the multiple batches are made, we think that these active components would in any case be the same. There would be a minor variation of active components.

  1. ln line 148, Why were doses of Rg3-RGE and PT altered when given in combination?
  2.  

Answer: Respected Reviewer, We thank you for your useful query. Since the main aim of our study is to check the synergistic effects of both extracts in combination; therefore, we had to select a dose that would cause less than 80% inhibition individually, so that when we give in combination, it would indicate either synergism or additive effect. Please view our raw data below for the individual NO inhibition for both extracts as follows:

  1. Markers for figure 5C are practically indistinguishable from each other. These need to be changed.

Answer: Respected Reviewer, We thank you for your comment. We have improved Figure 5C and added colours for ease of identification. We thank you again for your comments.

  1. Ln 158 ‘To investigate whether the in vitro results were due to colitis induction in mice’. What?

Answer: Respected Reviewer, We thank you for your useful concern over this query. We have rewritten this sentence for a better understanding please (Now line 163).

  1. Ln 176-177 ‘the mixture brings on signal propagation via NF-κB pathway.’ The mixture blocks or inhibits signal propagation through the NF-κB pathway?

Answer: Respected Reviewer, We thank you for your useful concern over this query. We have rewritten the sentence for a better understanding (Now line 185).

  1. Ln 193-195 for clarity give appropriate figure reference for each set of data.

Answer: Respected Reviewer, We thank you for your query. We have given the reference figure accordingly please (Now line 202-203).

  1. Ln 292 Give details of the DSS used in the study [product number, 40kDa?].

Answer: Respected Reviewer, We thank you for your concern. We have given the complete product information in the materials and methods section 4.1, line 305 please.

  1. Were daily water and DSS intakes monitored for the test and control groups?

Answer: Respected Reviewer, we thank you for your useful concern over this query Yes, the daily intakes were recorded for the fresh water and DSS water in mice. From our findings there were no significant difference in the water intake among the groups.

Respected reviewer, very humbly, we hope that we have answered to all the queries in a careful and detailed manner. We would again like pay a humble gratitude for reviewing our manuscript and giving useful comments for quality of manuscript.

With all due respect and regards,

Man Hee Rhee.

Reviewer 3 Report

The article by Saba E. et al. investigates separately and in combination the effects of Korean red ginseng extract enriched with Rg3 (Rg3-RGE) and Persicaria tinctoria (PT) in in vitro and in vivo experimental murine model in which ulcerative colitis (UC) was induced by dextran sulfate sodium (DSS).

They demonstrated that following the in vitro combined treatment on murine macrophage RAW 264.7, there was synergism in significant inhibition of NO production, as well as expression of iNOS, COX-2, IL-1β, IL-6 and TNF-α mRNAs. They demonstrated also that inhibition of inflammation was via mitogen-activated protein kinase and nuclear factor κ B pathways. Interestingly, they also showed that in C57BL/6 mice only the combined treatment was able to restore colon length, to hamper histopathological damage, and to decrease the levels of NLRP3 inflammasome.

They also studied the level of expression of pro-inflammatory mediators and cytokines (NO, IL-1β, IL-5, IL-13, and TNF-α) in vivo in mice following single or combined treatment and found that mixture group significantly inhibited these levels when compared to the DSS group.

Authors conclude by saying that the Rg3-RGE and PT mixture can be used for the prevention of UC as a prophylactic as well as therapeutic supplement.

Overall, the article is interesting and promising; it is well described and follows a consequential logical thread.

Although the results presented are of good relevance, however, there are some inaccuracies that need to be clarified.

1- Concerning the effects of Rg3-RGE + PT on immune cell subtypes in spleen, it is not clear why the authors characterize NK cells as CD4+CD69+, when there are specific markers for NK cells, such as NK1.1 for example. Indeed, NK cells are CD3-CD4-CD8-. Authors need to clarify this critical point.

2- Regarding experiments in which authors investigate the differentiation of T helper cells by examining secretion of IFN-γ related to Th1, and IL-4 related to Th2 in anti-CD3 stimulated splenocytes, one might argue that in whole splenocytes anti-CD3 stimulation could also induce cytokine secretion in CD3+ NKT cells. To conclude of the effects of combined treatment on Th1/Th2 rebalance it would be better to perform T cell negative purification.

3- At the same time the results obtained on the Th1/Th2 balance in my opinion are not conclusive of an incisive effect on the inhibited differentiation of Th0 towards Th1 and of an increase in Th0 towards Th2, as explained in the article. Rather, there could be a quantitative effect on the Th1 and Th2 final effectors, due to a decrease in IFNg production and an increase in IL-4.

Minor concerns:

Regarding M&M section Fluorescent antibody cell sorting (FACS), authors must detail the anti-CD mAbs and fluorochromes used and corresponding clones and controls, as well as describe the gating strategy performed.

Finally, in the article there are several errors concerning verbs that must be corrected and for example they are:

1- In the Abstract, line 23: The results obtained demonstrates…, instead of: The results obtained demonstrate.

2- In the Abstract, line 26: mixture exhibited strong anti-inflammatory effects…, instead of: mixture exhibits strong anti-inflammatory effects.

3- In the Abstract, line 28: Our results recommended…, instead of: Our results recommend…

4- In the Introduction, line 39: which in turn releases pro-inflammatory mediators and cytokines that aggravate inflammation, instead of: which in turn release pro-inflammatory mediators and cytokines that aggravate inflammation.

And so on.

Author Response

Reviewer 3

The article by Saba E. et al. investigates separately and in combination the effects of Korean red ginseng extract enriched with Rg3 (Rg3-RGE) and Persicaria tinctoria (PT) in in vitro and in vivo experimental murine model in which ulcerative colitis (UC) was induced by dextran sulfate sodium (DSS).

They demonstrated that following the in vitro combined treatment on murine macrophage RAW 264.7, there was synergism in significant inhibition of NO production, as well as expression of iNOS, COX-2, IL-1β, IL-6 and TNF-α mRNAs. They demonstrated also that inhibition of inflammation was via mitogen-activated protein kinase and nuclear factor κ B pathways. Interestingly, they also showed that in C57BL/6 mice only the combined treatment was able to restore colon length, to hamper histopathological damage, and to decrease the levels of NLRP3 inflammasome.

They also studied the level of expression of pro-inflammatory mediators and cytokines (NO, IL-1β, IL-5, IL-13, and TNF-α) in vivo in mice following single or combined treatment and found that mixture group significantly inhibited these levels when compared to the DSS group.

Authors conclude by saying that the Rg3-RGE and PT mixture can be used for the prevention of UC as a prophylactic as well as therapeutic supplement.

Overall, the article is interesting and promising; it is well described and follows a consequential logical thread.

Although the results presented are of good relevance, however, there are some inaccuracies that need to be clarified.

  1. Concerning the effects of Rg3-RGE + PT on immune cell subtypes in spleen, it is not clear why the authors characterize NK cells as CD4+CD69+, when there are specific markers for NK cells, such as NK1.1 for example. Indeed, NK cells are CD3-CD4-CD8-. Authors need to clarify this critical point.

Answer: Dear reviewer, we thank you for your suggestion. We agree with your point that NK 1.1 is the marker for NK cells. However in some studies CD69+ is also used for the identification of early, non specific NK cells marker, as reported by Fogel et. al.. A study by Marzio et. al. has also reported that CD69 is a member of the NK cell gene complex. Hence, we utilized CD4+CD69+ as NK cell marker in our study. We agree that in future studies we should use NK1.1 as it is a direct marker of NK cell in mice. The attached reference is as shown below:

  • Fogel LA, Sun MM, Geurs TL, Carayannopoulos LN, French AR. Markers of nonselective and specific NK cell activation. The Journal of Immunology 2013;190:6269-76.
  • Marzio R, Mauel J, Betz-Corradin S. CD69 And regulatiof the immune function. Immunopharmacology and immunotoxicology 1999;21:565-82.

  1. Regarding experiments in which authors investigate the differentiation of T helper cells by examining secretion of IFN-γ related to Th1, and IL-4 related to Th2 in anti-CD3 stimulated splenocytes, one might argue that in whole splenocytes anti-CD3 stimulation could also induce cytokine secretion in CD3+ NKT cells. To conclude of the effects of combined treatment on Th1/Th2 rebalance it would be better to perform T cell negative purification.

Answer: Dear reviewer, thank you for your suggestion. The insights you have provided is indeed valid. CD3+ NKT cells will secrete IFN-γ which may induce a false positive in detected IFN-γ levels. But in our study, the extracts have suppressed IFN-γ levels. Moreover, we are concerned that what when we incubate cells for differentiation in CD3 stimulation, T cells take time to differentiate, which may be difficult to gauge how long after CD3 stimulation that T cell negative purification should be conducted and from which time point onwards that the supernatant should be collected for detection of cytokines. We acknowledge this is a shortcoming of our study. Hence, we have added a statement in the manuscript stating this limitation in lines 211-213.

  1. At the same time the results obtained on the Th1/Th2 balance in my opinion are not conclusive of an incisive effect on the inhibited differentiation of Th0 towards Th1 and of an increase in Th0 towards Th2, as explained in the article. Rather, there could be a quantitative effect on the Th1 and Th2 final effectors, due to a decrease in IFNg production and an increase in IL-4.

Answer: Dear reviewer, we thank you for your suggestion. We agree that more validations should be conducted to come to a conclusion that the mixture directly acts on the Th1/Th2 balance. Therefore, we have altered our statements to mention that our findings may indicate that the mixture is probably acting through the Th1/Th2 balance. We also added that more studies should be conducted to validate this finding in lines 213-216. We have also altered the title of our manuscript to “Alleviation of ulcerative colitis potentially through Th1/Th2 cytokine balance by a mixture of Rg3-enriched Korean red ginseng extract and Persicaria tinctoria”. We thank you for identifying the shortcoming of our study and we appreciate it.

Minor concerns:

  1. Regarding M&M section Fluorescent antibody cell sorting (FACS), authors must detail the anti-CD mAbs and fluorochromes used and corresponding clones and controls, as well as describe the gating strategy performed.

Answer: Dear reviewer, we thank you for your suggestion. We have added the fluorochromes, stated their respective clones in the materials section, as shown in red in lines 310-313. In our study, we did not use isotype controls for determining negative population. We have blocked the cells with BSA and analyzed unstained cells in determining the negative population. We have also added that in our study we only gated lymphocytes for analysis in lines 400-401. These sections were added in lines. We thank you for your suggestion.

  1. Finally, in the article there are several errors concerning verbs that must be corrected and for example they are:
  2.  

2a. In the Abstract, line 23: The results obtained demonstrates…, instead of: The results obtained demonstrate.

Answer: Dear reviewer, thank you for your suggestion. We have made the changes necessary for line 24-25 in the abstract section, shown in red.

2b. In the Abstract, line 26: mixture exhibited strong anti-inflammatory effects…, instead of: mixture exhibits strong anti-inflammatory effects.

Answer: Dear reviewer, thank you for your suggestion. We have made the changes needed in line 27 in the abstract section as shown in red.

2c. In the Abstract, line 28: Our results recommended…, instead of: Our results recommend…

Answer: Dear reviewer, thank you for your suggestion, the sentence in line 29 has also been changed accordingly. We thank you for your suggestion.

2d. In the Introduction, line 39: which in turn releases pro-inflammatory mediators and cytokines that aggravate inflammation, instead of: which in turn release pro-inflammatory mediators and cytokines that aggravate inflammation.

And so on.

Answer: Dear respected reviewer, thank you for pointing this out to us. We have made the necessary changes needed in line 40-41, as shown in red.

Round 2

Reviewer 2 Report

The authors have dealt in a satisfactory manner with the queries raised. However, I believe that the paper would be strengthened by the addition of a paragraph/s in the discussion summarising the comments presented for query 1.

This would allow the reader to consider the nature and mode of action (additive or synergistic) of the known bioactive components in the two mixtures.

Author Response

The authors have dealt in a satisfactory manner with the queries raised. However, I believe that the paper would be strengthened by the addition of a paragraph/s in the discussion summarising the comments presented for query 1.

This would allow the reader to consider the nature and mode of action (additive or synergistic) of the known bioactive components in the two mixtures.

Dear reviewer, we thank you for your suggestion. We agree with your suggestion. We have added the relevant statements in a section in the discussion section, shown in red and highlighted in yellow from lines 262-274. We sincerely thank you again for your suggestion and your efforts in helping us improve our work.

Reviewer 3 Report

Concerning major point number 1 question, authors still need to modify/improve this critical point in the article.

Indeed, I have checked the two paper references proposed by authors, but mouse phenotype NK cells still need specific markers to identify them (i.e. NK1.1, NKp46...). In the Fogel LA et al paper (JI 2013;190:6269-76) NK cells have been identified as CD3neg NK1.1+, or as CD3neg, NK1.1+, NKp46+ (see materials and methods).

Moreover, NK cells are CD3neg CD4neg. Therefore, I would suggest to erase NK cells from the article. At the same time the % values of CD4+CD69+ cell subset were very low (between 1-2%).

Moreover, in the figure legend 8 it is not correct to put the % as absolute numbers. It should therefore be changed absolute with % number.

The other changes were made in accordance with the requests made.

Author Response

Reviewer 3

Concerning major point number 1 question, authors still need to modify/improve this critical point in the article.

Indeed, I have checked the two paper references proposed by authors, but mouse phenotype NK cells still need specific markers to identify them (i.e. NK1.1, NKp46...). In the Fogel LA et al paper (JI 2013;190:6269-76) NK cells have been identified as CD3neg NK1.1+, or as CD3neg, NK1.1+, NKp46+ (see materials and methods).

Moreover, NK cells are CD3neg CD4neg. Therefore, I would suggest to erase NK cells from the article. At the same time the % values of CD4+CD69+ cell subset were very low (between 1-2%).

Moreover, in the figure legend 8 it is not correct to put the % as absolute numbers. It should therefore be changed absolute with % number.

The other changes were made in accordance with the requests made.

Dear respected reviewer, we thank you for your suggestion and we agree with your statement. In the future, we will use NK 1.1 as a marker for NK cells. Hence, we have removed figure 8D and accordingly altered the figure legends and statements in the text. We thank you for your suggestion in helping us improve our work.

We have also altered the figure legend by replacing ‘absolute number’ into ‘percentage’, as shown in red highlighted in yellow.

We sincerely thank you for your efforts in reviewing our work.